Natural differential privacy—a perspective on protection guarantees

http://orcid.org/0000-0001-7382-6960 Altman Micah 1 micah_altman@alumni.brown.edu
Cohen Aloni 2
1 CREOS, MIT Libraries, Massachusetts Institute of Technology , Cambridge, MA , United States
2 Computer Science, University of Chicago , Chicago, IL , USA
Akleylek Sedat
Electronic publication date: 2023 Sep 28
Publication date: 2023
Volume: 9
Electronic Location ID: e1576
Received 2023 May 24; Accepted 2023 Aug 15
Copyright: © 2023 Altman and Cohen
Copyright year: 2023
Copyright holder: Altman and Cohen
License: This is an open access article distributed under the terms of the Creative Commons Attribution License, which permits unrestricted use, distribution, reproduction and adaptation in any medium and for any purpose provided that it is properly attributed. For attribution, the original author(s), title, publication source (PeerJ Computer Science) and either DOI or URL of the article must be cited.
License URL: https://creativecommons.org/licenses/by/4.0/

Keywords: Differential privacy, Physical mechanisms, No free lunch, Privacy by design, Privacy by default

Funding: The authors received no funding for this work.

==============================
We introduce “Natural” differential privacy (NDP)—which utilizes features of existing hardware architecture to implement differentially private computations. We show that NDP both guarantees strong bounds on privacy loss and constitutes a practical exception to no-free-lunch theorems on privacy. We describe how NDP can be efficiently implemented and how it aligns with recognized privacy principles and frameworks. We discuss the importance of formal protection guarantees and the relationship between formal and substantive protections.

Introduction

Differential privacy offers provable privacy guarantees (Wood et al., 2018) but has been criticized as difficult to integrate into existing data production systems and requiring substantial utility loss (Blanco-Justicia et al., 2022).

We introduce natural differential privacy (NDP)—a framework for guaranteeing differential privacy for arbitrary computations by leveraging features of existing hardware architectures and natural sources of entropy.

NDP provides all the advantages of “pure” differential privacy as originally formulated in Dwork et al. (2006), not resorting to any of the myriad relaxed definitions that provide weaker guarantees (Pejó & Desfontaines, 2022). NDP provides a worst-case bound on privacy loss that is substantially better than some high-profile, large-scale commercial implementations of DP.

Moreover, in contrast to existing implementations, NDP provides privacy by default for all computations on a platform. Furthermore, NDP is simple and inexpensive to implement at a large scale and requires no practical reduction in utility or performance.

Organization

The article proceeds as follows: Section 3 provides a definition of NDP, explains the concept, and provides theoretical foundations, including provable privacy guarantees. Section 4 discusses related work and explains how NDP differs from other approaches. Section 5 discusses the advantages of NDP in comparison to other approaches and how NDP aligns with existing privacy principles and frameworks. Section 6 discusses implementation approaches—including sufficient hardware and software features for implementation—and analyzes the privacy parameter obtained (and thus protection guaranteed) in exemplar implementations. Section 7 summarizes experimental results. Section 8 characterizes limitations and directions for future research—including the relation to other security properties. Section 9 provides a more general discussion—informed by NDP—of the substantive importance of formal privacy guarantees and guidance for the interpretation of protection claims.

Concept: preliminaries, definition, abstract implementation

We consider the setting of a dataset x consisting of n records, where each record is a bitstring of dimension d. We view each row as containing the data of a single individual. Databases x and x′ are neighboring if they differ in at most one record. A mechanism M is a randomized mapping from datasets to some set of possible outputs Y.

Definition 2.1 (ε-Differential Privacy (ε-DP) (Dwork et al., 2006)). M is ε-differentially private if for all neighboring datasets x and x′, and for all sets S ⊆ Y:

(1) Pr[M(x)∈S]≤eϵ⋅Pr[M(x′)∈S]

where the probabilities are taken over M’s coins.

NDP is defined as any system that integrates DP protections directly in a Von Neumann architecture (von Neumann, 1945) via hardware implementation using persistent sources of entropy for noise infusion. By construction, every internal computation by an NDP system integrates noise infusion guaranteeing (ε, 0)-DP. And because DP preserves privacy under postprocessing—all outputs from the system are thus (ε, 0)-DP.

Implementation: NDP applies to the computation of arbitrary m-bit functions f of the data x, for any m. To evaluate the NDP-version of f, one simply evaluates f on a RAM machine. For best results, the RAM should be operated at or above sea level.

We use as a building block the Randomized Response mechanism. The Randomized Response mechanism is parameterized by a probability 0 < p < 0.5, and we denote the corresponding mechanism RRp. RRp takes as input a bit b ∈ {0,1}, and outputs 1-b with probability p, otherwise outputting b. Results established in Willis (2014) provide a formula for the exact equivalence between the probability of randomized response and the epsilon parameter. For any p < 0.5, Eq. (2) expresses this relationship:

(2) 1−p=exp(ϵ)/(1+exp(ϵ))

Privacy parameters: Let Tin > 0 and Tout > 0 be the length of time that the input x and output f(x) are stored in RAM over the course of the computation, respectively.1

Each computation has a corresponding parameter q that depends on the environment within which the computation is performed. Thus q is the probability of any single bit flip caused by cosmic rays occurring on 1 GB of RAM over the course of 1 day (see Table 1). From q, it is easy to derive the probability p that any single bit is flipped in the period Tin or Tout. Using p and applying Eq. (1), it is straightforward to solve for a value of ε (at the bit event level).2

Table 1 Privacy budget configuration through altitudinal adjustment.

Derivative-free numerical minimization (Brent, 1973) is used to obtain epsilon corresponding to p, given Eq. (2). Bit-level frequency data is provided by Soft Error (2022), Enyinna (2016). Epsilon levels are calculated for protection at the (bit) event-level, To calculate epsilon for other units of protection, it is straightforward to calculate the effective epsilon by using the standard dp composition formula across the maximum number of shared events in the computation (Kairouz, Oh & Viswanath, 2015). Even under composition, the effective epsilon remains trivially small relative to the baseline: Where B is the number of independent bits measured per unique event, the protection produced by this method is, in the worst case, εB.

m > sea-level	Exemplar location	μSv/h	Error/GBxDay	Max ε	
−3,840	Mponeng gold mine	0	0	∞	
10	Cambridge, MA	0.06	0.2	33.70711	
1,742	Mount Wilson observatory	0.237	2	31.39832	
10,000	Jet airplane’s lower cruising altitude	6	60	27.99537	
781,000	Iridium satellite constellation	60	600	25.69307	

Related work

Other variants of DP, such as epsilon-delta DP and concentrated DP, have been proposed (Dwork & Rothblum, 2016).3 However, these variants relax the definition of DP yielding weaker privacy properties (We refer hereafter to such relaxations as ‘artificial’).

Current implementations of DP at scale have used artificial DP. Because of the substantial utility tradeoffs that artificial DP often requires—commercial implementations often use values of epsilon well above 1. Recent large-scale implementations of differential privacy by major corporations (including Google and Apple) have employed effective epsilon levels ranging from dozens to hundreds—with one major implementation exceeding seven hundred and fifty4 (Rogers et al., 2020).

Natural sources of entropy for noise diffusion have been studied for over four decades (Ziegler & Lanford, 1979). Their importance in security and privacy has been recognized in related areas: Bit flipping induced by radiation or other environmental conditions has been previously used for practical attacks against system security (Dinaburg, 2011; Govindavajhala & Appel, 2003).

The importance of high-quality random number generation for all differential privacy methods has recently been recognized (Garfinkel & Leclerc, 2020). Nearly all implementations rely on pseudo-random sequences seeded from a physical entropy source. The use of physical sources of randomness for direct noise infusion has not been well-examined.

More recently, the inherent instability of quantum computation has been examined as a theoretical source of protective noise infusion—although practical implementation remains far off (Zhou & Ying, 2017).

Advantages of ndp

Although natural noise infusion has been studied in related work, the use of natural sources directly for differential privacy is novel. The NDP approach offers a number of advantages: NDP protects all computations made on a system.

NDP does not require any relaxation of the formal differential privacy guarantees—unlike artificial DP variants.

NDP is simple to implement and inexpensive to deploy.

NDP provides protections that are substantively equivalent (or better) than the formal guarantees provided by notable commercial implementations—while maintaining higher utility, substantially reducing implementation cost, and extending protection to a broader range of computations.

Further, DP has additional attractive features: First, NDP encourages privacy by design (Cavoukian, 2009)—NDP can be integrated into hardware architecture, systems design, and facility deployment, as well as at the application level.

Second, NDP provides privacy by default (Willis, 2014) since a floor for protection is provided for all users without requiring any opt-ins (NISO, 2015).

Third, NDP aligns well with the widely adopted ‘five safes framework’ (Desai, Ritchie & Welpton, 2016). Specifically, it uses architectural privacy by design to guarantee ‘safe outputs’.

Fourth, NDP can be implemented either at the time of manufacture or during deployment. This facilitates certification and auditing of secure hardware and facilities.

Fifth, NDP provides guaranteed, measurable privacy with zero marginal utility loss—theoretical no-free-lunch theorems notwithstanding (Kifer & Machanavajjhala, 2011).5

Setting privacy parameters within ndp systems—hardware and software implementation

NDP, when fully integrated into the architecture, operates by default, continuously, and at the hardware level. In modern architectures, which implement random access memory using MOSFET technology, bit-flipping is an ideal mechanism for noise infusion.

At manufacturing time, the infusion of readily available alpha sources into memory chip packaging material requires no increase in manufacturing cost. This approach can be used to ensure a maximum bound for epsilon that is both auditable through inspection and can be verified through hardware certification processes.6

More generally, ionizing radiation provides a natural and ubiquitous source for inducing randomized responses at the bit level—acting through the injection of memory faults. Utilizing this physical mechanism for entropy provides true randomness, which provides stronger formal guarantees than pseudorandom number generation (Vadhan, 2012; Garfinkel & Leclerc, 2020).

Results established in Wang, Wu & Hu (2016) provide a formula for the exact equivalence between the probability of randomized response and the epsilon parameter.

(3) p=1/(1+exp(ϵ))

Many sources of radiation are readily available, as shown in Fig. 1.7

Figure 1 Comparison of readily available entropy sources for setting ε.

(Source Munro (2011), in the public domain).

Post-manufacture, maximum bounds on epsilon can be further reduced at deployment time through the positioning of the hardware. Entire computing facilities may be certified as safe using this method.

Conveniently, cosmic rays cause i.i.d. bit flips at a rate directly related to the level of cosmic radiation (typically measured in micro-sieverts) exposure, which is itself a function of atmospheric density (above sea level) and crustal density (below sea level) at a given altitude. The levels of radiation experienced at specified altitude relative to sea level has been established empirically and fits a geometric distribution within near-earth orbiting distances. Using these empirical results and Eq. (3), we can derive the effective value of epsilon at exemplar locations, as shown in Table 1.

Note that relative to baseline, very low values of epsilon can be achieved through altitudinal adjustment. Further, note that the level of epsilon provided naturally at sea level is more protective than the level provided by the most notable and largest scale production implementations of differential privacy to date (Greenberg, 2017). Finally, in practice, the effective epsilon will be statistically indistinguishable from implementations using a theoretically lower value.8

At runtime, noise injection can readily and effectively be achieved by altering the thermal operating environment (Govindavajhala & Appel, 2003). Further, in high-density computing deployments, simply reducing the level of external cooling will not only increase protective noise infusion but also reduce electricity usage—benefitting the global environment. Moreover, various external noise injection tools are available and can provide additional protection without affecting the location or manufacturing process (Hsueh, Tsai & Iyer, 1997). Finally, given the wide availability of (level-2) hypervisor-based system-level virtual machine technologies, simulation-based noise infusion (aka. synthetic natural differential privacy) can be used to produce any desired level of epsilon with a relatively small decrease in runtime performance (Li et al., 2017).

Experimental results

The primary contribution of the article is a formal analysis of NDP properties. And the derivation of protection levels for a specific implementation rely on the value of physical parameters (such as levels of background radiation and the corresponding bit-level error rates) that are already well-established in the literature (as detailed in the previous section).

For replication, we conducted a set of implementation experiments deployed in the exemplar locations in Table 1, Cambridge, MA. The experiment involved running 25 output audits of the form described above in footnote 9. The experiment was conducted on a 12th Gen Intel Core i5-1235U CPU, using LPDDR4 RAM, and running the Linux operating system (See Appendix for replication code). The expected protection level was observed in all 25 experiments, replicating previously established values at conventional levels of significance (p = 0.05).

Limitations and future work

On occasion, serious points are best conveyed through humor.9 This article is intended as a work of (serious) humor. While each of the individual technical assertions in the article is true, catastrophic drawbacks are omitted or glossed over. Thus the substantial benefits claimed for the method, particularly in “Advantages of NDP,” are parodically misleading.

For example, some of the limitations of the above proposal include the following: Formal privacy-protecting methods aim to limit adversarial inferences based on outputs. Other potentially desirable security properties, such as integrity, non-repudiation, etc., are orthogonal to this goal. Thus it is often appropriate to use privacy-protecting methods in conjunction with other privacy-enhancing technology and security controls. See OECD (2023), Altman et al. (2015) for selected reviews of complementary approaches.

As is true generally for formal approaches to privacy, guarantees are provided under the assumption that the adversary does not observe timing characteristics of the computation, which would potentially allow for timing channel attacks (Biswas, Ghosal & Nagaraja, 2017).

As is generally true for privacy-preserving methods, guarantees rely on the assumption that the computing platform itself is secure. For example, if adversaries can intercept input data prior to applying the protection method, observe all computations on the system, or alter the implementation of arbitrary code, including protecting algorithms, this may subvert privacy and other security properties. For these reasons, privacy-protecting methods are often deployed in a secure environment to produce outputs that may then be shared publicly—and complementary security controls are deployed with privacy-protecting methods. For a review of complementary security controls recommended in conjunction with privacy protective methods generally and with DP-related methods specifically, see Altman et al. (2015), Wood et al. (2020).

Use of ε above 1, although frequent in practice, requires caution for any DP-based method.

It is rarely the proper objective of law or public policy to protect the privacy of an ‘event’. It is usually a more appropriate policy goal to protect the privacy of a persistent actor—such as a person, organization, or designated group of people.

Information about an event or other unit of protection is rarely limited to a single bit. Where measurements of the unit of protection comprise multiple bits, composition will multiplicatively increase the effective ε. Further multi-bit events should be encoded efficiently—so that each bit of input data represents a unique measured bit of information (This limitation applies only to inputs. Other representations, such as floating-point, may be used for computations on those inputs. See footnote 2).

Randomized response does not generally result in an efficient tradeoff between privacy and utility—especially as the effective information measured grows large.

Inducing a high level of bit flipping at random in main memory may cause unwanted side effects—such as entirely incorrect results, nonsense output, program failures, and system crashes.

Thus, there is no market for highly unreliable RAM, despite the low cost of manufacture.

Further, introducing ionizing radiation to the deployment site to achieve a meaningful level of protection may risk exposure to lethal amounts of radiation and create a durable hazard. This potentially violates local regulations, national laws, and international treaties.

Discussion

One of the triumphs of modern computer science is the ability to establish formal provable protection guarantees. However, when using any real system, it is vital to understand the extent, strength, and conditions under which provable protections.

The limitations above notwithstanding, there is a sense in which cosmic rays formally induce ε-DP. Further, the epsilon values used by some large-scale commercial implementations of DP provide provable worst-case privacy-loss guarantees that are arguably no stronger than those provided by cosmic rays in Cambridge.

Increasingly, production systems claim the benefit of provable privacy protections—almost all of them based on some form of differential privacy. When, as often happens, very high values of epsilon are employed, the amount of protection that is provable, while technically measurable, is substantively negligible.

This article, while parodic, illustrates that provable formal privacy and truthful compliance with privacy principles neither guarantee substantive protection nor require substantial implementation effort. Provable guarantees, such as those provided by differential privacy, have force only when the specific level of protection provided by implementation privacy parameters in practice is meaningful, the guarantees apply to a well-defined and substantively meaningful set of interactions between protected units and protective systems, and when the formal unit of protection corresponds to real-world entities with meaningful privacy interests. Moreover, when a system embeds a weak implementation of a protection mechanism at its core, compliance with other privacy principles, such as privacy-by-design, may offer little value.

Protections may still be of substantial value even if they are not formally proven—these implementations may provide useful protection in particular contexts even if such protections are not formally provable.10 However, such protections should be established on their own merits—using appropriate non-formal (and especially, empirical) evidence. Notwithstanding, when the strong guarantees provided by a system are substantively negligible, it is grossly misleading to claim the system provides provable protection.

Supplemental Information

Supplemental Information 1 R Code For Replication.

Click here for additional data file.

We thank Stephen Chong and Kobbi Nissim for their helpful comments on prior drafts. We describe the authors’ contributions following a standard taxonomy (Allen et al., 2014).

Additional Information and Declarations

Competing Interests

Author Contributions

Data Availability

1 If the portions of the input (respectively, output) are in RAM for different lengths of time, then we take Tin (resp., Tout to be the minimum time over every bit of the input (resp., output).

2 Many consider the event-level as the most practical unit of protection for streaming systems (Lécuyer et al., 2019)—and has been used at scale for public release of large scale Facebook data for scientific research (King & Persily, 2019, 2020) In addition, the value of ε for multi-bit events is arithmetically derivable—see Footnote 8. Note that as with any other random-response based approach, redundant, or compressed encoding of measurements prior to protection may result in under- or over-protection. Specifically, input data should be represented as bit-fields, integers or character strings in which each bit of the data corresponds to a bit of information being protected. For example, integers, characters, or bit-fields as implemented in C++ (ISO, 2020) would be suitable. The induced value of ε for inputs stored in floating point representations (IEEE Computer Society, 2019) remains a topic of research. However, computations on the input data are unrestricted—thus protected functions may use IEEE standard floating point computations as long the initial input data are represented as integers (etc.).

3 Differential privacy is the most developed and popular of a larger set of possible formal privacy measures—for a survey of current measures see Wagner & Eckhoff (2018). Further the core protection provided by formal privacy measures is to limit (in a statistically well-defined way) what an adversary can infer from protected outputs. In practice, formal approaches may be complemented by a range of other privacy-enhancing technologies (e.g., that provide limits on computation rather than inference)—as well as complementary security control mechanisms. The scope of complementary technologies is vast and expanding—so a discussion is beyond the scope of this article. For reviews of potential complementary privacy technologies see OECD (2023), Altman et al. (2018).

4 Estimates are for the effective protection of user information over a month of activity. Because of composition effects, the effective epsilon for protection of a user information in these systems grows geometrically over time. Thus a cumulative epsilons of ten thousands or more is possible for frequent, long-term users of these systems.

5 In theory, theory and practice are the same—in practice, they differ. See Brewster (1882), Quote Investigator (2018) for the original formulation by Brewster, and contributions attributed to Berra and Einstein.

6 Even without changes in manufacturing processes, manufacturers can readily certify that their RAM is not unsafe by design—in that it does not provide ECC or this feature has been disabled (e.g., at the firmware level).

7 This chart illustrates approximate point estimates of source intensity, for use in general comparison of types and magnitude of sources. For specific details on each source see Mettler et al. (2008), Munro (2023).

8 Assuming a continuous audit period of 1 h, and conventional levels of statistical significance (p = 0.05) is used, the observed value at sea level will not be statistically distinguishable from a theoretical epsilon of 31.91007 (Zero bits flipped will be observed during that period at either level of epsilon, during at least 95% of audits).

9 See Janko (1987) for foundations of this principle, and Van Stempvoot, Portinga & Johnson (2022) for both a modern defense and exemplar of the principle.

10 And it is always possible that as new methods of formal analysis are developed, we may come to understand that some systems are more protective than could initially be proved.

The authors declare that they have no competing interests.

Micah Altman conceived and designed the experiments, contributed to the conception of the report (including core ideas and statement of research questions), to the methodology, and to the writing through critical review and commentary, performed the experiments, analyzed the data, performed the computation work, prepared figures and/or tables, authored or reviewed drafts of the article, and approved the final draft.

Aloni Cohen conceived and designed the experiments, contributed to the conception of the report (including core ideas and statement of research questions), to the methodology, and to the writing through critical review and commentary, authored or reviewed drafts of the article, and approved the final draft.

The following information was supplied regarding data availability:

The replication code is available in the Supplemental File.

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
