# Peer review of "Natural differential privacy—a perspective on protection guarantees"

_PeerJ Computer Science, doi:10.7717/peerj-cs.1576_

## Round 0.1 · original submission · Major Revisions

The referral process is now complete. While finding your paper interesting and worthy of publication, the referees and I feel that more work should be done before the paper is published. My decision is therefore to provisionally accept your paper subject to major revisions.

Reviewer 1 ·

Basic reporting

The research paper titled "Natural Differential Privacy (NDP): Utilizing Hardware Architecture for Differentially Private Computations" introduces an interesting concept that leverages existing hardware architecture to implement differentially private computations. While the paper addresses important aspects of privacy guarantees and the practicality of NDP, there are several areas that can be improved to enhance the research:

Clarity of the Concept: The paper should provide a clear and concise explanation of the concept of Natural Differential Privacy (NDP) and how it differs from existing approaches. It should outline the specific features of hardware architecture that enable the implementation of differentially private computations. This will help readers understand the novelty and significance of NDP.

Theoretical Foundation: The research paper should provide a strong theoretical foundation for NDP. It should discuss the mathematical framework behind differential privacy and how NDP aligns with recognized privacy principles and frameworks. This would help establish the credibility and rigor of the proposed approach.

Experimental Validation: To validate the effectiveness and practicality of NDP, the paper should include experimental evaluations or simulations. This would demonstrate the performance of NDP in terms of privacy guarantees and efficiency compared to existing methods. It would also help assess the scalability and applicability of NDP to real-world scenarios.

Comparison with Existing Approaches: The paper should provide a comprehensive comparison of NDP with existing approaches for implementing differentially private computations. This includes discussing the strengths and weaknesses of NDP in relation to other methods. Additionally, the paper should highlight the specific scenarios or use cases where NDP outperforms alternative approaches.

Practical Implementation Details: The paper should provide detailed information on how NDP can be efficiently implemented on different hardware architectures. This includes describing the specific hardware features utilized and any modifications required. Providing implementation guidelines or code examples would be valuable for researchers and practitioners interested in adopting NDP.

Privacy and Security Analysis: The paper should thoroughly analyze the privacy guarantees and security properties of NDP. This includes assessing the potential vulnerabilities or attack vectors that could compromise privacy in the NDP implementation. Addressing these concerns and discussing potential countermeasures would enhance the reliability and trustworthiness of NDP.

Discussion and Future Directions: The research paper should include a comprehensive discussion of the implications and limitations of NDP. It should explore potential future research directions to further improve the concept and address any remaining challenges. This could involve addressing scalability issues, considering different types of hardware architectures, or exploring the integration of NDP with other privacy-enhancing technologies.

By addressing these areas for improvement, the research paper on Natural Differential Privacy (NDP) can strengthen its contribution, validity, and practicality. This will make the paper more valuable to the research community and provide a solid foundation for future advancements in differentially private computations using hardware architecture.

Experimental design

see Basic reporting

Validity of the findings

see Basic reporting

Cite this review as

Reviewer 2 ·

Basic reporting

This study parodically criticizes commecial implementations of differential privacy as a protection mechanism, arguing that DP usage with malconfigured parameters offers very little and even "natural" resources of bit flipping event (if not prevented) would be much more stronger.

The paper is very well written, and quite fun to follow. The reviewer would like to admit that this is the first instance of this flavor of scientific work that he has gone through and thanks the authors for this opportunity.

Here are some points to consider:
* A clearer discussion on the composition properties of NDP would be helpful. This subject is verbally discussed in footnote 6.
* Line 149 correctly states that NDP noise is statistically indistinguishable from implementations using a theoretically lower value. Isn't that the case for PRNGs as well?

Apart from these, I have nothing to add.

Experimental design

All the formulae in the manuscript is correct. There are no experiments, just factual information found in Figure 1 and Table 1.

Validity of the findings

The code submitted together with manuscript is sufficient to replicate the facts given in the paper.

Cite this review as

Reviewer 3 ·

Basic reporting

The authors are using natural radiation, which can be the cause of a bit-flip to preserve the privacy of the data. It can be an exciting approach for mimicking differential privacy. However, the author also stated that the bitflips are random and occur in any part of the IEEE-754 64-bit representation. Therefore, it can also dramatically damage the overall computation, which is deeply investigated in the literature. This situation is one of the significant drawbacks of the proposed approach, as also mentioned in the study's limitations. Another possibility is if the bitflip occurs in the least significant part of the mantissa, there will be no visible change at the floating point number. Therefore, there should be a logic to check the bitflips to ensure the safety of computation and the privacy of the data simultaneously. Without such logic, the proposed method has no applicability. As stated by the authors in the limitations part of the study, implementing this approach needs many assumptions. Although the initial idea can be considered as promising, it needs more advanced research before it deserves to be published.

Experimental design

no comment

Validity of the findings

no comment

Additional comments

Some of the references need to be removed or revised. The authors should avoid using XKCD cartoons as an informative figure in their paper.

Cite this review as

---

## Round 0.2 · Minor Revisions

The reviewers have pointed out some minor issues.

Reviewer 1 ·

Basic reporting

Authors updated the paper and no further update needed from my side.

Experimental design

see Basic reporting

Validity of the findings

see Basic reporting

Additional comments

see Basic reporting

Cite this review as

Reviewer 2 ·

Basic reporting

The reviewer would like to thank the authors for answering all concerns of the review from the previous.

Here are some comments on the newly added material:
- Footnote 2: "remains a topic OF research".
- Footnote 3: security control families --> security control mechanisms.
- Footnote 7: "This chart illustrateS". Formatting should agree with other footnotes.
- Sec. 7, 2nd sentence: "derivation OF protection levels".
- The last sentence of Sec. 8 may be omitted altogether, it stands out.

Obviously, these are very minor corrections.

Experimental design

Same as previous round: All the formulae in the manuscript are correct. There are no experiments, just factual information found in Figure 1 and Table 1.

Validity of the findings

Same as previous round: The code submitted together with manuscript is sufficient to replicate the facts given in the paper.

Additional comments

None.

Cite this review as

Reviewer 3 ·

Basic reporting

The new version of the manuscript and the answers of the authors regarding the reviewer comments are satisfactory. Therefore, the manuscript can be published.

Experimental design

No extra comments.

Validity of the findings

No extra comments.

Cite this review as

---

## Round 0.3 · accepted · Accept

Since the reviewers' comments have been answered, we are happy to inform you that your manuscript has been accepted for publication.